# The Diagnostic Value of EEG Wave Trains for Distinguishing Immature Absence Seizures and Sleep Spindles: Evidence from the WAG/Rij Rat Model

**DOI:** 10.3390/diagnostics15080983

**Published:** 2025-04-12

**Authors:** Olga S. Sushkova, Alexei A. Morozov, Alexandra V. Gabova, Karine Yu. Sarkisova

**Affiliations:** 1Kotel’nikov Institute of Radio Engineering and Electronics of RAS, Mokhovaya St. 11-7, 125009 Moscow, Russia; morozov@cplire.ru; 2Institute of Higher Nervous Activity and Neurophysiology of RAS, Butlerova St. 5A, 117485 Moscow, Russia; agabova@yandex.ru (A.V.G.); karine.online@yandex.ru (K.Y.S.)

**Keywords:** absence epilepsy, animal model, WAG/Rij rat, wave-train electrical activity, immature absence seizure, sleep spindle

## Abstract

**Background:** Absence epilepsy is a non-convulsive form of genetic generalized epilepsy characterized by spontaneous bilateral spike-and-wave discharges (SWDs) in EEG. In contrast to grand-mal epilepsy, absence epilepsy without greatly expressed motor and interictal EEG abnormalities is difficult to detect, especially at the early stages. The WAG/Rij rat strain is a well-validated animal model of childhood absence epilepsy. At the early, preclinical stage, precursors or immature SWDs appear. Then, with age, immature discharges gradually turn into mature ones and mature SWDs prevail at the clinical stage. Mature SWDs, with an amplitude several times higher than the background EEG, can be easily distinguished visually. However, the amplitude of immature discharges is significantly lower than that of mature SWDs and is comparable to the amplitude of sleep spindles. Therefore, it is quite a difficult problem to distinguish immature discharges from sleep spindles. The task is further complicated by the fact that absence seizures mainly appear in a state of drowsiness and slow-wave (non-REM) sleep, when a lot of sleep spindles occur. The purpose of the present study was to develop a diagnostic method that allows us to precisely distinguish immature forms of epileptic seizures from background EEG and sleep spindles. **Methods:** The idea of analyzing wave-train electrical activity is to investigate the wavelet spectrum, find local peculiarities in this spectrum, and estimate generalized time-frequency peculiarities of the signal in terms of the found local peculiarities. **Results:** The criteria for diagnosis of the immature form of epileptic discharges and sleep spindles have been developed based on the analysis of wave-train activity with the construction of AUC diagrams (area under the curve diagrams). **Conclusions:** The method of wave-train analysis with the construction of AUC diagrams can be used for extracting the diagnostic features necessary for the diagnosis of absence epilepsy at the early stages of the disease in people with a genetic predisposition.

## 1. Introduction

Absence epilepsy is a non-convulsive form of genetic generalized epilepsy characterized by the presence of spontaneous, synchronous, and symmetric spike-and-wave discharges (SWDs) in EEG accompanied by a loss of consciousness (absences) [1]. Typical absences usually start in childhood or adolescence, but they can occur also in around 10 to 15% of adults with epilepsy, often combined with other generalized seizures, in particular, with tonic–clonic seizures [2,3].

The most common effective diagnostic method for epilepsy detection is based on the analysis of EEG signals. EEG analysis allows for not only distinguishing epileptic seizures from non-epileptic EEG events but also differentiating ictal (epileptic) signals from pre-ictal and post-ictal ones [4,5]. Unlike grand-mal seizures, absence seizures are difficult to detect. Interictal (background) EEG readings are usually normal (without visible EEG abnormalities), which makes it difficult to diagnose non-convulsive absence epilepsy [6]. Moreover, absence seizures are usually so brief that they frequently escape notice. Conversely, it is much easier to diagnose the convulsive temporal lobe epilepsy with well-marked paroxysmal spikes on the interictal EEG. The possibility of using the characteristics of interictal spike activity to predict the severity of future seizures and the outcome of surgery is being investigated [7].

Additional atypical (focal) interictal EEG features can occur along with features typical for absence epilepsy, which may result in misdiagnosis or delay diagnosis [6,8]. Moreover, patients with typical absences may also exhibit interictal epileptiform discharges that have similar features of the ictal pattern, creating difficulties for differentiation and diagnosis [9,10]. Pre-existing typical absences that were undiagnosed in childhood can aggravate and lead to a non-convulsive absence status in adults under the influence of exogenous triggers [11]. Older age of onset is one of the early predictors of additional generalized tonic–clonic seizures in absence epilepsy [12]. These data suggest that accurate early diagnosis is essential for the prognosis of disease development and adequate therapy.

Animal models of epileptic seizures are very important for the investigation of the pathophysiological mechanisms underlying epileptogenesis, as well as for the development of new methods for early diagnosis and new preventive treatment strategies. Animal models of absence epilepsy have been shown that correct, early treatment not only suppresses seizure occurrence in later life but can also prevent the development of associated depression-like comorbidity [13,14].

The WAG/Rij rat line is one of the well-tested animal models of childhood absence epilepsy [15,16,17,18,19]. The first well-developed (mature) SWDs, which are characteristic of typical absence seizures, appear in WAG/Rij rats at the age of 2–3 months, and then their number and duration increase with age. At the early, preclinical stage of disease development, the precursors of SWDs (immature discharges) are present as early as 2 months of age. Then, with age, immature discharges undergo three stages of “maturation” [20,21] and mature SWDs prevail at the clinical stage with a fully manifested pathology (around the age of 6–7 months) typical for absence epilepsy. Mature SWDs, with a relatively abrupt onset and offset and an amplitude several times higher than the background EEG, can be easily distinguished visually. However, the amplitude of immature discharges is significantly lower than that of mature SWDs and is comparable to the amplitude of sleep spindles. Therefore, it is quite a difficult problem to distinguish immature discharges from sleep spindles. The problem is further complicated by the fact that absence seizures mainly appear in a state of drowsiness and slow-wave (non-REM) sleep when a lot of sleep spindles occur. The purpose of the present study was to develop a diagnostic method that allows us to sufficiently distinguish immature forms of epileptic seizures from background EEG and sleep spindles.

Currently, wavelet analysis is a generally accepted tool for analyzing EEG signals [20,22,23,24,25,26,27,28,29,30,31]. Methods for detecting epileptiform activity and sleep spindles have been developed on the basis of wavelet analysis [22,23,24,25,26,27,28,29,30,31,32]. It is shown that the complex Morlet wavelet is a good choice for EEG analysis [32,33]; however, the windowed fast Fourier transform is also used [34,35]. The authors [32,36] formalize the description of epileptiform activity and sleep spindles based on the study of wavelet spectrograms of EEG signals and develop an algorithm for the automatic detection of EEG sections corresponding to this description. The disadvantage of this approach is that the features of epileptiform activity and sleep spindles are formulated on the basis of ideas of the expert but not on the results of statistical analysis of the data, which can lead to the loss of useful information about the properties of EEG signals. In most other works [22,23,24,25,26,27,28,29,30,31,37,38], automatic extraction of the features of epileptiform activity and sleep spindles is implemented using neural networks, random forests, support vector machine (SVM), and other machine learning methods. The automatic approach allows for extracting more information about the properties of EEG signals; however, the EEG signal analysis algorithm becomes a black box, which complicates the neurophysiological interpretation of the identified features of EEG signals. The method of analyzing wave-train electrical activity (WTEA) is a compromise between the two approaches because the search for regularities in EEG signals is carried out using various statistical tools and the identified regularities are visually represented in the form of graphic diagrams, which facilitates their neurophysiological interpretation.

The idea of analyzing WTEA is to investigate a wavelet spectrum, find local peculiarities in this spectrum, and estimate generalized time-frequency peculiarities of the signal in terms of the found local peculiarities [39,40,41]. The method is founded on the use of so-called AUC diagrams [39,40,41]. The reasons for developing a new method for EEG analysis are as follows. According to the authors, the weakness of the existing methods for biomedical signal analysis is that they focus on determining the global characteristics of the signal over long periods of time, as in Fourier analysis, or on determining the local time-frequency characteristics of the signal, as in wavelet analysis [42,43,44,45,46]. As a result, the existing signal analysis methods miss a large amount of useful information, including global characteristics of local signal properties.

This paper applies a method of analyzing the brain’s WTEA to develop criteria for differential diagnostics of immature epileptic discharges and sleep spindles. The method is described using an example of EEG data analysis of WAG/Rij rats. The second part of this paper describes the experimental dataset. The third part of this paper describes the EEG data acquisition procedure. The fourth part of this paper describes the method for analyzing EEG data. The fifth part of this paper describes the results of comparing immature discharges and background EEG. The sixth part describes the results of comparing immature discharges and sleep spindles. The seventh part of this paper provides a correlation analysis of the number of blue and red wave trains that distinguish immature discharges, sleep spindles, and background EEG.

## 2. Animals

Nine 6-7-month-old males of inbred WAG/Rij rats were used in this study. WAG/Rij rats were bred and raised at the Institute of Higher Nervous Activity and Neurophysiology of the Russian Academy of Sciences (IHNA and NPh RAS). The rats were kept in a temperature-controlled vivarium under a 12 h light–dark cycle (lights were turned on at 8.00 a.m.). Animals were housed in plexiglas cages (3–4 animals in a cage). Food (standard rat chow) and drinking water were available ad libitum. After the implantation of electrodes for EEG recordings, the rats were individually housed. The care and use of animals were ensured by the institutional policies and guidelines. The experimental protocols were approved by the IHNA and NPh RAS Ethics Committee (approval code no. 5 of 2 December 2020). Every effort has been made to minimize the number of animals used for research and their suffering from pain according to the 3R concept.

## 3. EEG Registration

Stereotactic surgery was performed under chloral hydrate anesthesia (400 mg/kg, i.p.). The cortical EEG electrodes (small stainless steel screws) were implanted epidurally over the frontal cortex (2 mm anterior and 2.5 mm lateral to bregma) and occipital cortex (6 mm posterior and 4 mm lateral to bregma). The electrodes were placed in small circular holes (0.8 mm diameter) in the skull and fixed with dental acrylic. The reference electrode was placed over the cerebellum. The ends of the wires coming from the screw electrodes were soldered to a 5-pin connector. The soldering points were covered with dental acrylic. The rats were allowed to recover from the implantation of the EEG electrodes for at least 7 days. Thereafter, the animals were placed in plexiglas recording cages (width × length × height was 15 × 30 × 26 cm) and adapted to the experimental conditions within 1 h before the start of the recording session. To register the EEG, a 5-pin connector located on the rat’s skull was fitted with an outlet part of the connector with a flexible cable (Figure 1). EEG signals were fed into a multi-channel amplifier via flexible cable attached to the outlet connector, band pass filtered between 1 and 40 Hz, digitized with a 500 Hz sampling rate per channel, and transmitted to the PC via Wi-Fi. All the EEG recordings were stored on a hard disk for further off-line analysis.

The EEG was recorded monopolarly. The EEG recordings were performed for 3 h per day (from 16.00 to 19.00 h) in freely moving animals. The custom-made wireless 8-channels biopotential measurement system BR9V1 (A. Troshchenko, IHNA and NPh RAS, Moscow, Russia) based on the Texas Instruments ADS1298 Analog Front-End was used.

The SWDs and sleep spindles recorded in the frontal cortex, in which they have the largest amplitude and a more pronounced morphological structure compared with the occipital cortex, were analyzed. The EEG recordings from the frontal cortex of the left hemisphere (the F1 electrode) were used for analysis.

The samples of mature (epileptic) and immature (not fully developed, pre-epileptic) discharges and sleep spindles were highlighted by an experienced expert. Mature and immature discharges were taken for analysis from EEG recordings during wakefulness; sleep spindles were taken from EEG recordings during slow-wave (non-REM) sleep. Mature SWDs and immature discharges were first selected visually. Then, the visual detection of SWDs, immature discharges, and sleep spindles was verified using wavelet and Fourier analysis and morphological features as previously described [47]. Mature SWDs were discovered in EEG as repetitive sequences of sharp, asymmetric, and high-amplitude spikes and slow waves (spike–wave complexes) lasting more than 2 s, with an amplitude at least three times higher than the background EEG signal (Figure 2a). The mature SWD is composed exclusively of the sequence of spike–wave (SW) complexes (Figure 2b). The wavelet spectrogram reflected the time-frequency characteristics of the mature SWD: the epileptic discharge began at a higher frequency of 11–12 Hz; then the frequency decreased rapidly, within about 0.5 s; and, further, the discharge had a stable frequency of 7–8 Hz (Figure 3). The Fourier power spectral density (PSD) of the mature SWD had the first harmonic at a fundamental frequency (7.5 Hz) with the second and third harmonics at the frequencies of 14.5 and 21.5 Hz, respectively (Figure 4).

The immature discharge was differentiated from the mature one by the irregular appearance of a spike–wave sequence with lower amplitude and nearly symmetrical deflection (Figure 5A, 1). The wavelet spectrogram showed an unstable frequency of the immature discharge (Figure 5A, 2). The Fourier power spectrum was disorganized and had no pronounced fundamental frequency and harmonics (Figure 5A, 3). Unlike the mature SWD, the immature discharge was composed of an SW sequence intermingled with wave-like oscillations (Figure 5A, 4). Sleep spindles in a slow-wave (non-REM) sleep were defined as a sequence of symmetrical wave-like complexes with a predominant frequency of 7–14 Hz and the amplitude exceeded the background activity by 2 times (Figure 5B, 1). The wavelet spectrogram of the sleep spindle showed an unstable frequency (Figure 5B, 2) and had no fundamental frequency and harmonics (Figure 5B, 3) similar to the immature discharge. Sleep spindles of epileptic WAG/Rij rats often contained sharp waves (S) resembling spikes (Figure 5B, 4).

## 4. EEG Analysis Based on Wave Trains

For the data analysis, special software was developed in the Matlab language. The preliminary processing of the EEG signals was carried out:The 50 and 100 Hz notch filters were applied to remove a power-line noise.The filtering of the EEG with an eighth-order Butterworth filter with a passband from 0.1 to 120 Hz was applied in the forward and reverse directions.

The spectrograms were calculated using the complex Morlet wavelet (Equation 1):(1)$$\psi \left(x\right)=\frac{1}{\sqrt{\pi F_{b}}}exp\left(2\pi ıF_{c}x\right)exp\left(\frac{-x^{2}}{F_{b}}\right)$$
where $F_{b}$ = 1 and $F_{c}$ = 1. The frequency step was 0.1 Hz. Using wavelet analysis, the EEG signal is transformed into a set of local maxima in the wavelet spectrogram, from which local maxima that meet certain conditions are taken for analysis (see Appendix A).

Figure 6 and Figure 7 demonstrate an instance of an EEG wave train and its wavelet spectrogram.

We considered the following wave-train characteristics: the central wave-train frequency, the maximum PSD of the wave train, the wave-train duration in seconds (at $1/\sqrt{2}$ of its height), the wave-train duration in periods (at $1/\sqrt{2}$ of its height), the wave-train bandwidth (at $1/\sqrt{2}$ of its height), and the instantaneous phase of the wave train. These characteristics form a certain multidimensional space. From a mathematical point of view, the WTEA method is aimed at finding a certain subspace *S* in which differences in the number of wave trains between the studied groups of data are observed.

We use AUC diagrams [39,40,41] for the comparison of data groups. Each wave-train characteristic corresponds to a special type of AUC diagram. For instance, the frequency AUC diagram shows the degree of difference between two data groups depending on the selected frequency range in the EEG signal (see Figure 8). Each cell in the frequency AUC diagram refers to a specific frequency band. The horizontal axis of this cell corresponds to the lower limit of this band and the vertical axis of this cell corresponds to the upper limit of the band. The degree of difference between the data groups is estimated by the AUC (area under ROC curve [48]), which is indicated by color. See the details on calculating ROC curves in Appendix A.

The frequency band from 0.1 to 50 Hz is considered in the frequency AUC diagram in Figure 8. The frequency step is of 0.1 Hz. The lightjet colormap is used to display the AUC value, that is, the red color denotes an AUC close to 1 and the blue color denotes an AUC close to 0. AUC values close to 1 means that the number of wave trains in immature discharges is greater than in the background EEG. AUC values close to 0 means that the number of wave trains in immature discharges is less than in the background EEG. For the sake of brevity, from now, we will call the wave trains that are typical for immature discharges as red wave trains and wave trains that are typical for background EEG and/or sleep spindles as blue wave trains.

To find the subspace *S* of the wave-train characteristic space in which differences in the number of wave trains between the studied data groups are observed, we used a genetic optimization algorithm [49].

## 5. Comparison of Immature Discharges and Background EEG

Based on the analysis results, it was discovered that wave trains that distinguish immature discharges from the background EEG have a PSD of 50,000 μV^2^/Hz and higher. The frequency AUC diagram computed using the specified constraint on the wave-train parameters is demonstrated in Figure 9. Note that no blue areas are observed in Figure 9 in contrast to Figure 8. Thus, we were able to separate wave trains typical for immature discharges and wave trains typical for the background EEG as a result of refining the wave-train parameters. The results found using the genetic optimization algorithm are confirmed by the AUC diagram of the PSD (see Appendix A).

Figure 10 demonstrates a histogram of the number of red wave trains. The histogram of the number of red wave trains observed in immature discharges is drawn in purple and the histogram of the number of red wave trains observed in the background EEG is drawn in green. Note that the purple and green histograms are well separated. This result is an obvious consequence of the fact that the amplitude of immature discharges significantly exceeds the amplitude of the background EEG.

In the frequency AUC diagram (Figure 8), in addition to the magenta area discussed above, there is also a dark blue area. This fact indicates that there are wave trains in the wavelet spectra of the background EEG that are not typical for immature discharges. In other words, the AUC diagrams show the presence of two features that distinguish groups of data. One feature is the presence of some wave trains in the immature discharges that are not typical for the background EEG (the red area in Figure 8), and the second feature, conversely, is the presence of some wave trains in the background EEG that are not typical for the immature discharges (the blue area in Figure 8). We performed an analysis of the blue area in Figure 8 using a genetic optimization algorithm.

The analysis discovered that the wave trains that discern background EEG from immature discharges have a frequency bandwidth of 2 Hz and higher. The frequency AUC diagram constructed using the specified constraint on the wave-train parameters is shown in Figure 11. Note that in Figure 11, the size and brightness of the red area have significantly decreased in comparison with the frequency diagram in Figure 8. Thus, we were able to separate wave trains typical for the background EEG and wave trains typical for immature discharges as a result of refining the wave-train parameters. The results found using the genetic optimization algorithm are confirmed by the AUC diagram of the frequency bandwidth (see Appendix A).

Figure 12 demonstrates a histogram of the number of blue wave trains. The histogram of the number of blue wave trains observed in immature discharges is drawn in purple and the histogram of the number of blue wave trains observed in the background EEG is drawn in green. The purple and green histograms are well separated. The neurophysiological interpretation of this result is not trivial. A wide frequency band is typical for signals of complex shape and noise-type signals. Thus, we can state that there are some wave trains with a wide frequency band in the background EEG, the number of which significantly decreases during an immature discharge.

We have analyzed the blue wave trains detected in the background EEG. For this purpose, we constructed histograms of the number of blue wave trains with a bandwidth of 2 Hz and higher observed in sleep spindles (see Figure 12, the blue histogram). It turned out that the number of blue wave trains decreases compared to the background EEG both in sleep spindles (the Mann–Whitney test, *p* < 2.43 $\times \;10^{-5}$ ) and in immature discharges (the Mann–Whitney test, *p* < 4.11 $\times \;10^{-5}$). At the same time, no significant difference was found between the number of blue wave trains in immature discharges and sleep spindles. This means that the observed effect of reducing the number of blue wave trains is not specific to immature discharges.

A possible explanation for the observed neurophysiological regularity is that there is a synchronous decrease in the membrane potential of large groups of neurons when sleep spindles and/or epileptiform activity occurs in the cerebral cortex; this leads to a decrease in the probability of spontaneous neuron excitation manifested in the form of noise-type EEG signals.

Table 1 contains the results of the analysis of Figure 8. The number of wave trains differs significantly in immature discharges and in the background EEG (Mann–Whitney test; *p*-values are given in Table 1).

See Appendix A for examples of wave trains corresponding to the blue and red areas in the AUC diagram comparing immature discharges (nine) with the background EEG (nine).

## 6. Comparison of Immature Discharges and Sleep Spindles

Immature discharges and sleep spindles are very similar [32,47]; thus, comparison of immature discharges and sleep spindles using the method of WTEA analysis is an urgent research problem. Figure 13 demonstrates a frequency AUC diagram comparing immature discharges with sleep spindles. The diagram does not contain pronounced monochromatic areas, which indicates the difficulty of the problem of distinguishing immature discharges and sleep spindles.

The genetic algorithm revealed the following differences between the parameters of wave trains typical for immature discharges and sleep spindles. Wave trains that distinguish immature discharges from sleep spindles have a central frequency of no more than 8.5 Hz, a duration of at least 1.6 periods, a bandwidth of at least 1.7 Hz, and an instantaneous phase of −π up to +1.5 radians. Note that a comparison of immature discharges and sleep spindles did not reveal any differences in PSD of wave trains typical for these signals. Thus, wave trains typical for the above signals differ mainly in frequency, duration, and shape.

The frequency AUC diagram computed using the given constraints on the wave-train parameters is shown in Figure 14. Compared to the frequency diagram in Figure 13, Figure 14 contains a brighter red region. Appendix A shows corresponding AUC diagrams of the wave-train duration in periods, wave-train bandwidths, and wave-train instantaneous phases.

Note that the parameters of the red wave trains that distinguish immature discharges from sleep spindles (central frequency, duration, bandwidth, and instantaneous phase) indicate that these wave trains are distinguished by their shape but not by their amplitude. Figure 15 demonstrates a histogram of the number of red wave trains. The histogram of the number of red wave trains observed in immature discharges is drawn in purple and the histogram of the number of red wave trains observed in sleep spindles is drawn in light blue. Note that the purple and light blue histograms are very close. This indicates that the shape of immature discharges differs significantly from the shape of sleep spindles, although in some cases it may be not possible to distinguish an immature discharge from a sleep spindle.

A similar analysis was performed for blue wave trains. A frequency AUC diagram (Figure 13) does not indicate any distinct dark blue regions. However, the genetic algorithm revealed the presence of blue wave trains that are typical for sleep spindles and not typical for immature discharges. The analysis discovered that the wave trains that discern sleep spindles from immature discharges have the following parameters: a central frequency of no more than 15 Hz, a bandwidth of at least 4 Hz, and an instantaneous phase from −2.5 to +2.5 radians. The frequency AUC diagram computed using the above constraints on the wave-train parameters is demonstrated in Figure 16. Note that a large blue area has appeared in Figure 16 in contrast to the frequency AUC diagram in Figure 13. Thus, we are able to separate the wave trains typical for sleep spindles from the wave trains typical for immature discharges with an AUC of approximately 0.1 as a result of refining the wave-train parameters. Appendix A contains the corresponding AUC diagrams of the wave-train bandwidth and the instantaneous phase of the wave trains.

Note that the parameters of the blue wave trains that distinguish immature discharges from sleep spindles (central frequency, bandwidth, and instantaneous phase) indicate that these wave trains are distinguished by their shape but not by their amplitude. Figure 17 demonstrates a histogram of the number of blue wave trains. The histogram of the number of wave trains observed in immature discharges is drawn in purple and the histogram of the number of wave trains observed in sleep spindles is drawn in light blue. The purple and blue histograms intersect. This confirms the conclusion that the shape of immature discharges differs significantly from the shape of sleep spindles, although in some cases it is not possible to distinguish an immature discharge from a sleep spindle.

Table 2 contains the results of the analysis of Figure 13. The number of wave trains differs significantly in immature discharges and sleep spindles (Mann–Whitney test; *p*-values are indicated in Table 2).

See Appendix A for examples of wave trains corresponding to the blue and red areas in the AUC diagram comparing immature discharges (9) with sleep spindles (20).

## 7. Discussion

The results of the comparison of immature discharges with the background EEG, as well as the comparison of immature discharges with sleep spindles, indicate that in both cases blue and red wave trains (that is, wave trains that are typical for or, conversely, not typical for immature discharges) are observed. At the same time, we can distinguish immature discharges and the background EEG in almost 100% of cases. Sleep spindles also differ significantly from immature discharges (in 83–85% of cases). However, we were unable to completely separate these groups of signals. It can be assumed that this may be due to the presence of similar components in the morphological structure of immature epileptic discharge and sleep spindles (Figure 5, 4). The fact that both blue and red wave trains were present in the signals under study required additional analysis. We conducted a correlation analysis of the number of wave trains in different groups of EEG signals to verify the presence of a cause-and-effect relationship between an increase in the number of red and a decrease in the number of blue wave trains.

The correlation analysis of the number of wave trains distinguishing immature discharges from the background EEG (see Table 1) revealed the following regularities. The Spearman correlation between the number of blue and red wave trains in immature discharges is 0.72 (the Spearman test *p* < 0.03). A significant Spearman correlation was also found between the number of blue and red wave trains in the background EEG. However, this correlation was negative: −0.85 (the Spearman test *p* < 0.006). Figure 18 demonstrates a scatter plot of the number of blue and red wave trains distinguishing immature discharges from the background EEG. Each point in the scatter plot designates one EEG fragment. The purple diamonds designate immature discharges. The green circles designate the background EEG fragments. *X* designates the number of wave trains per second in the red areas. *Y* designates the number of wave trains per second in the blue areas.

In the scatter plot (Figure 18), the EEG fragments corresponding to immature discharges and the background EEG fragments are well separated. The elongated shape of the purple and green clusters in the scatter plot confirms the presence of a positive and negative correlation between the number of blue and red wave trains in the considered groups of EEG fragments. This observation suggests that the number of blue and red wave trains is influenced by a single neurophysiological mechanism. However, in immature discharges and background EEG, the correlation between the number of blue and red wave trains changes dramatically. Two possible hypotheses can be proposed to explain this pattern. The first hypothesis is that the number of wave trains in immature discharges and background EEG is influenced by some different neurophysiological mechanisms. The second hypothesis is that the number of blue and red wave trains in immature discharges and background EEG is influenced by the same neurophysiological mechanism, but the operation of this mechanism changes.

The correlation analysis of the number of blue and red wave trains distinguishing immature discharges from sleep spindles (see Table 2) did not reveal any correlation between the number of blue and red wave trains in either immature discharges or sleep spindles. Figure 19 demonstrates a scatter plot of the number of blue and red wave trains distinguishing immature discharges from sleep spindles. Each point in the scatter plot designates one EEG fragment. Purple diamonds designate immature discharges. Blue squares designate sleep spindles. *X* designates the number of wave trains per second in the red areas. *Y* designates the number of wave trains per second in the blue areas.

In the scatter plot (Figure 19), the EEG fragments corresponding to immature discharges and sleep spindles are well separated. At the same time, the purple and blue clusters in the scatter plot partially intersect. All blue squares lie near *Y*, which indicates that immature discharges contain wave trains that are practically absent in sleep spindles. Thus, we observe a qualitative difference between immature discharges and sleep spindles. The number of blue and red wave trains in EEG fragments can serve as a reliable feature distinguishing these types of EEG signals. This is good news for researchers studying immature EEG discharges because it opens up prospects for standardizing procedures for automatic recognition of immature discharges, thereby reducing the influence of subjective factors on the interpretation of the results of neurophysiological experiments.

## 8. Conclusions

Despite the existence of a large number of EEG analysis methods, including methods for detecting and predicting absence seizures [50,51], there are relatively few methods for distinguishing absence seizures and sleep spindles [33]. As far as we know, there are practically no methods for the differential diagnosis of immature epileptic (pre-epileptic) activity and sleep spindles.

In the present study, a method has been developed to identify the features of the immature form of epileptic activity and sleep spindles in the EEG of WAG/Rij rats, a genetic model of human absence epilepsy. Criteria have been formulated that can be used for the automatic recognition of immature epileptic activity and sleep spindles, which may have translational significance for clinical use. The wave-train EEG analysis developed on the WAG/Rij rat absence model can be used to analyze human EEG for the early diagnosis of epilepsy. It should be noted that the method of EEG wave-train analysis has also been successfully applied in the clinic for the differential diagnosis of early stage of Parkinson’s disease and essential tremor [39].

The proposed method is founded on the use of wavelet spectrograms and AUC diagrams. An analysis of experimental EEG data using the developed method made it possible to identify features that distinguish immature discharges from background EEG and sleep spindles with high accuracy. For example, using the detected features in the algorithm (see patent number RU 2781622 C1 in Section 9) allowed us to achieve the specificity and sensitivity of recognizing immature discharges from sleep spindles of 0.95 and 0.89, respectively; the specificity and sensitivity of recognition of immature discharges against the background EEG is 100%.

The advantage of the developed method for detecting features of epileptic activity, in comparison with existing methods, is that the method is based on the search for regularities in EEG signals but not on models of epileptic activity. In addition, the method provides the experimenter with quantitative characteristics of EEG signals (the number of blue and red wave trains), which allows for comparing immature discharges with the background EEG and sleep spindles and making assumptions about the neurophysiological mechanisms underlying the studied electrical activity of the brain. The developed method opens up new prospects for solving the problem of the early diagnosis of absence epilepsy in people with a genetic predisposition to this disease.

## 9. Patents

Sushkova O.S., Morozov A.A., Gabova A.V., and Sarkisova K.Yu. Patent number RU 2781622 C1. Russian Federation. Method for detecting immature discharges in epilepsy in laboratory rats using the analysis of wave-train electrical activity of the brain. Published: 17 October 2022 Bul. No. 29. Application: 2022115970, 14 June 2022.

## Figures and Tables

**Figure 1 diagnostics-15-00983-f001:**
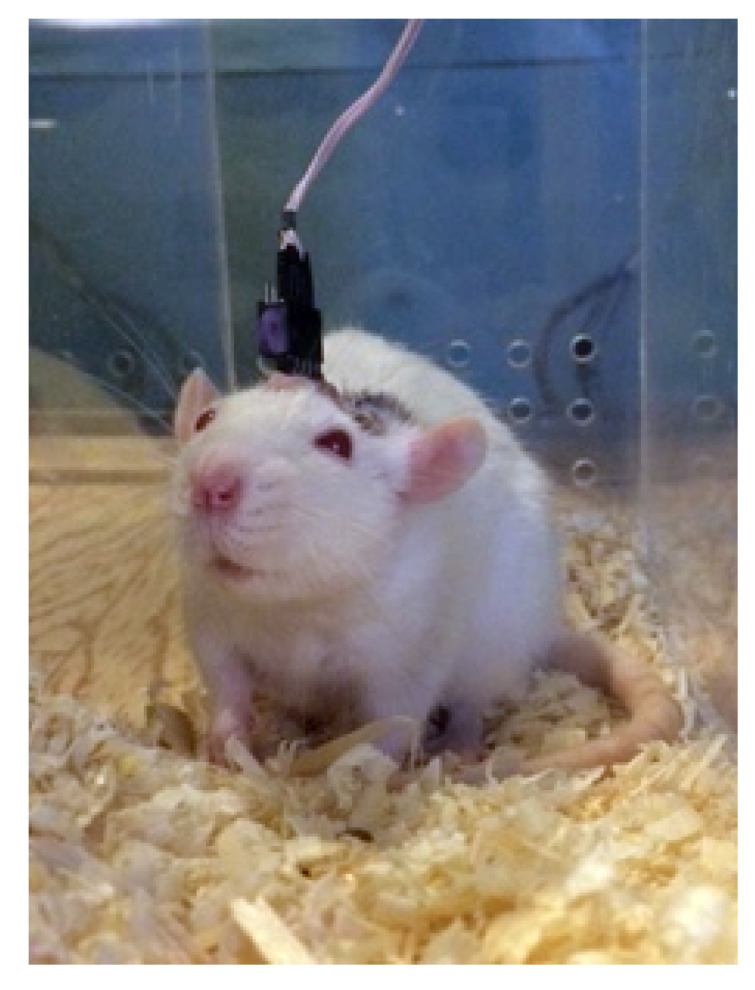
A WAG/Rij rat with a 5-pin connector located on the rat’s skull.

**Figure 2 diagnostics-15-00983-f002:**
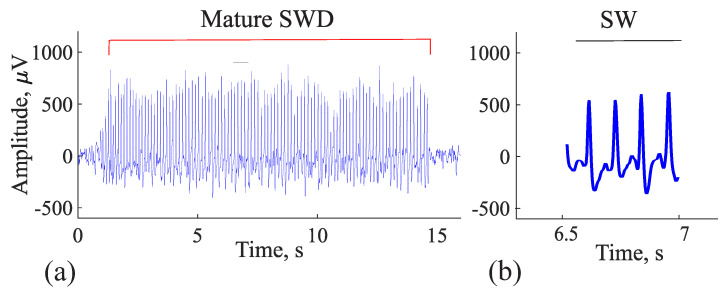
(**a**) An example of an EEG fragment with a mature SWD marked with a red horizontal line above the discharge. (**b**) A fragment of an SWD (marked by a black horizontal line above the mature discharge) is presented on an extended timeline to illustrate its morphology: a sequence of repetitive spike–wave complexes (SW). *X* designates the time, seconds; *y* designates the amplitude, μV.

**Figure 3 diagnostics-15-00983-f003:**
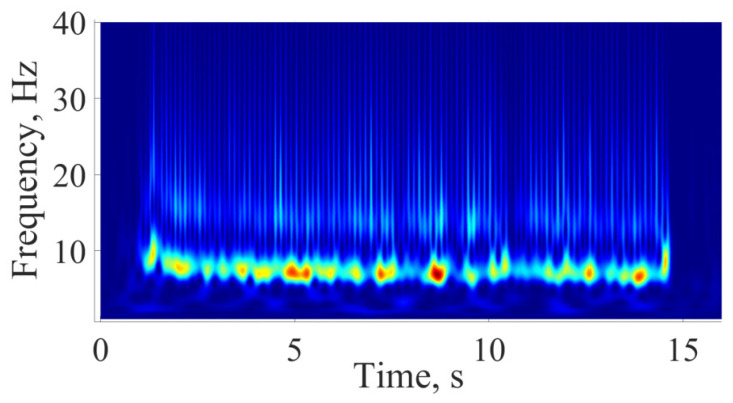
A wavelet spectrogram of an SWD reflecting its time-frequency dynamics. *X* designates the time, seconds; *y* designates the frequency, Hz. PSD is designated by color: blue indicates low PSD; red indicates high PSD.

**Figure 4 diagnostics-15-00983-f004:**
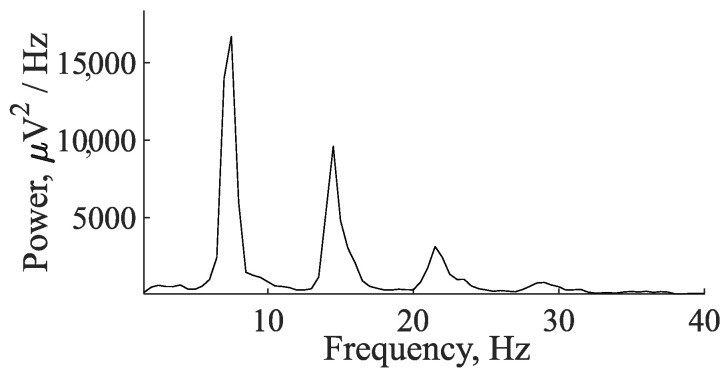
PSD of the SWD obtained by the fast Fourier transform using the Welch method. *X* designates the frequency, Hz; *y* designates the PSD, μV^2^/Hz.

**Figure 5 diagnostics-15-00983-f005:**
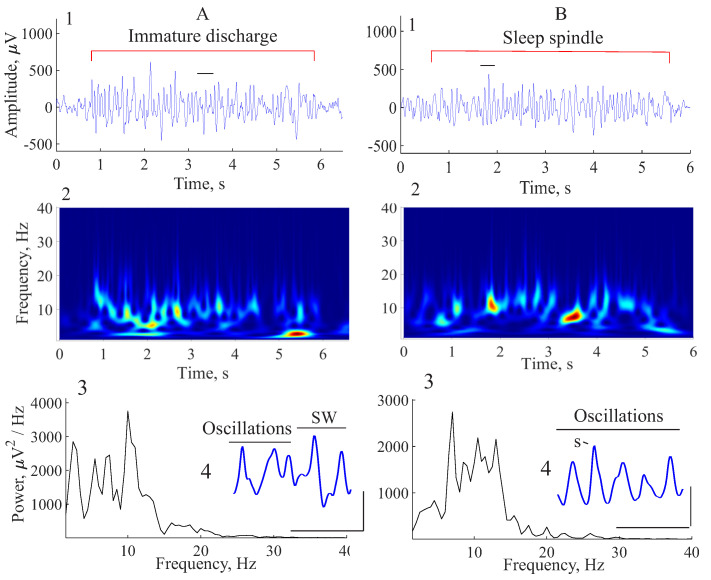
Characteristics of immature discharge (**A**) and sleep spindle (**B**). The immature discharge and sleep spindle are marked with red lines. (1) Examples of EEG of immature discharge and sleep spindle. *X* designates the time, seconds; *y* designates the amplitude, μV. (2) Wavelet spectrograms of immature discharge and sleep spindle. *X* designates the time, seconds; *y* designates the frequency, Hz. PSD is designated by color: blue indicates low PSD; red indicates high PSD. (3) PSD of immature discharge and sleep spindle. *X* designates the frequency, Hz; *y* designates PSD, μV^2^/Hz. (4) Fragments of the immature discharge and sleep spindle marked by a black horizontal line in A1 and B1 to illustrate their morphology; spike–wave complexes (SW) interspersed with wave-like oscillations (**A**, 4) and wave-form oscillations (**B**, 4) comprising sharp (spike-like) waves (S). Scale bars designate the following: time 500 ms (abscissa) and amplitude 500 μV (ordinate).

**Figure 6 diagnostics-15-00983-f006:**
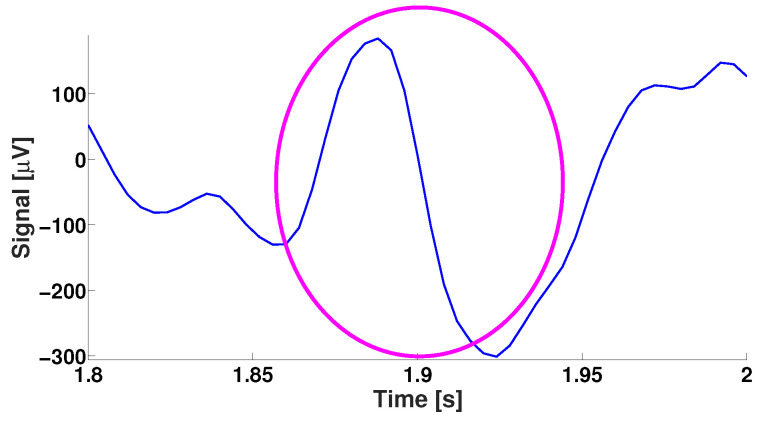
An instance of a wave train in an EEG signal in the rat. *X* designates the time in seconds. *Y* designates the amplitude in μV. The wave train is designated by a purple ellipse.

**Figure 7 diagnostics-15-00983-f007:**
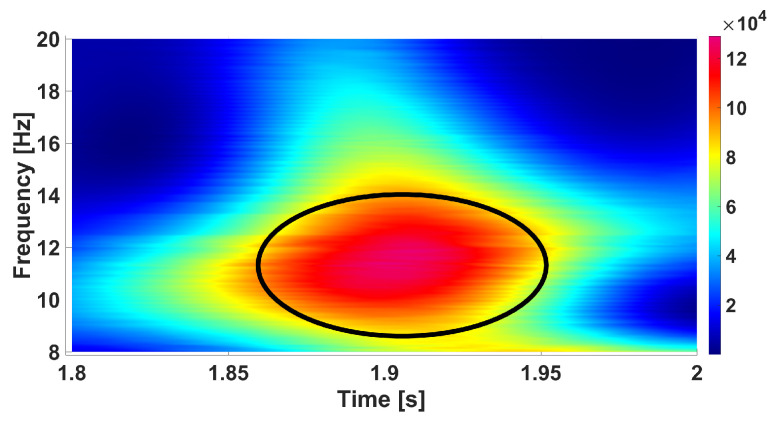
An instance of a wavelet spectrogram of the wave train in an EEG signal in the rat. *X* designates the time in seconds. *Y* designates the frequency in Hz. The wave train is designated by a black ellipse.

**Figure 8 diagnostics-15-00983-f008:**
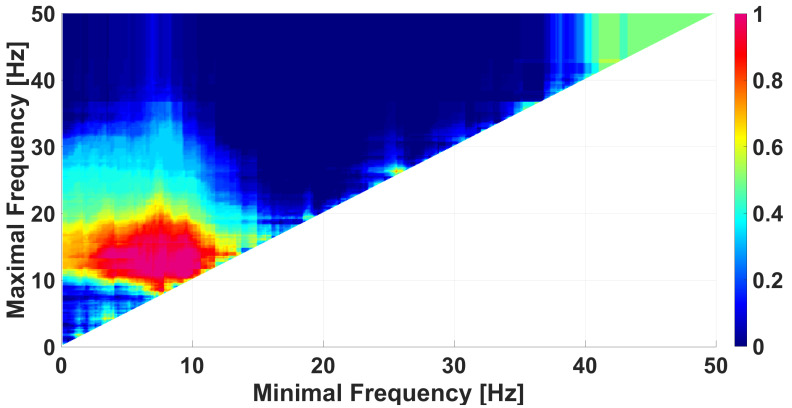
A frequency AUC diagram comparing immature discharges (9) with background EEG (9). *X* designates the lower limit of the frequency band; *y* designates the upper limit of the frequency band. The diagram contains one bright red area and one dark blue area.

**Figure 9 diagnostics-15-00983-f009:**
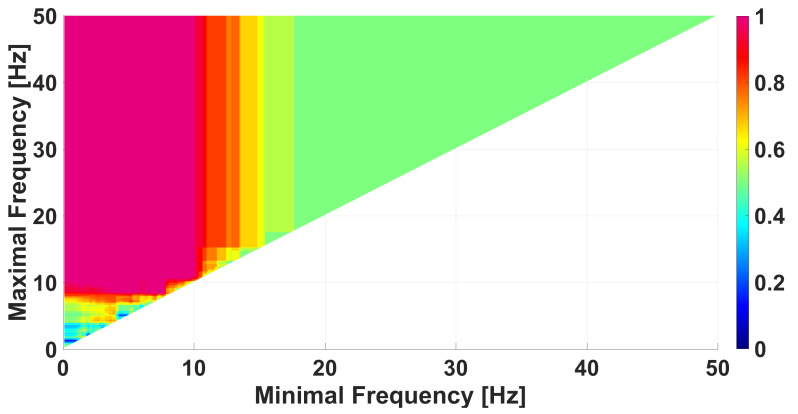
A frequency AUC diagram comparing immature discharges (9) with background EEG (9). The diagram is constructed using the following constraint on the wave-train parameters: PSD from 50,000 μV^2^/Hz and higher. *X* designates the lower limit of the frequency band; *y* designates the upper limit of the frequency band. The diagram contains one red area. The blue area is absent.

**Figure 10 diagnostics-15-00983-f010:**
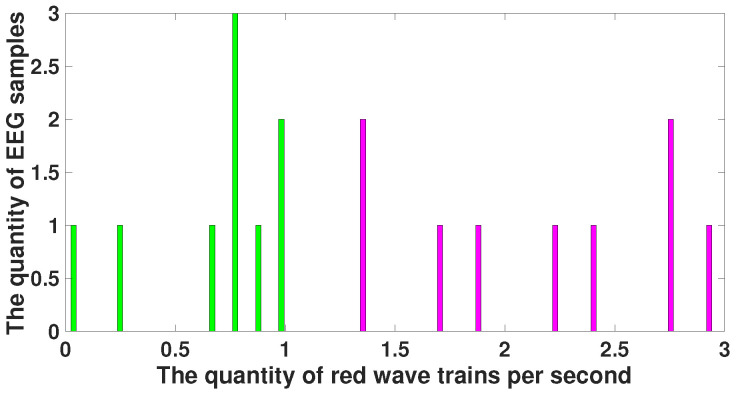
A histogram of the number of red wave trains. *X* designates the number of red wave trains per second; *y* designates the number of EEG fragments. The histogram of the number of wave trains observed in immature discharges is drawn in purple; the histogram of the number of wave trains observed in the background EEG is drawn in green. The purple and green histograms are well separated.

**Figure 11 diagnostics-15-00983-f011:**
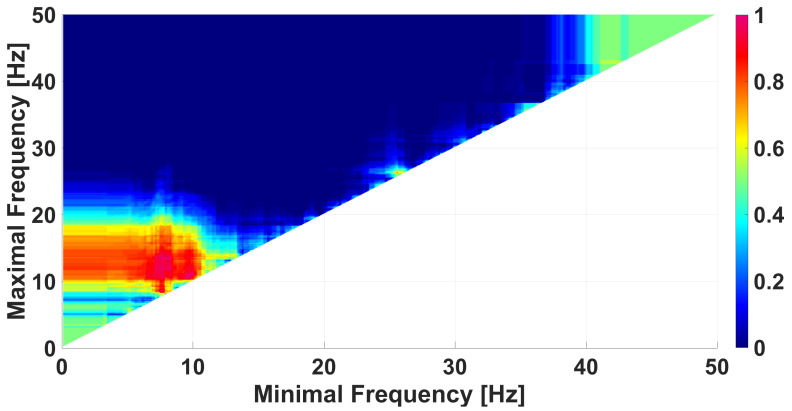
A frequency AUC diagram comparing immature discharges (9) with background EEG (9). The diagram is constructed using the following constraint on the wave-train parameters: bandwidth from 2 Hz and higher. *X* designates the lower limit of the frequency band; *y* designates the upper limit of the frequency band.

**Figure 12 diagnostics-15-00983-f012:**
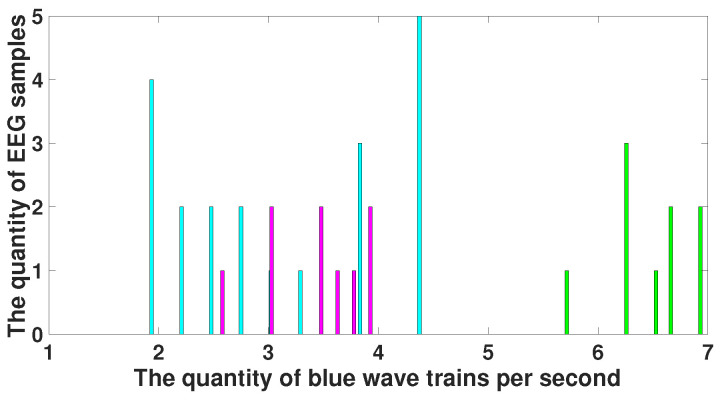
A histogram of the number of blue wave trains. *X* designates the number of blue wave trains per second; *y* designates the number of EEG fragments. The histogram of the number of wave trains observed in immature discharges is drawn in purple; the histogram of the number of wave trains observed in the background EEG is drawn in green; and the histogram of the number of blue wave trains observed in sleep spindles is drawn in light blue. The purple and green histograms, as well as the light blue and green histograms, are well separated.

**Figure 13 diagnostics-15-00983-f013:**
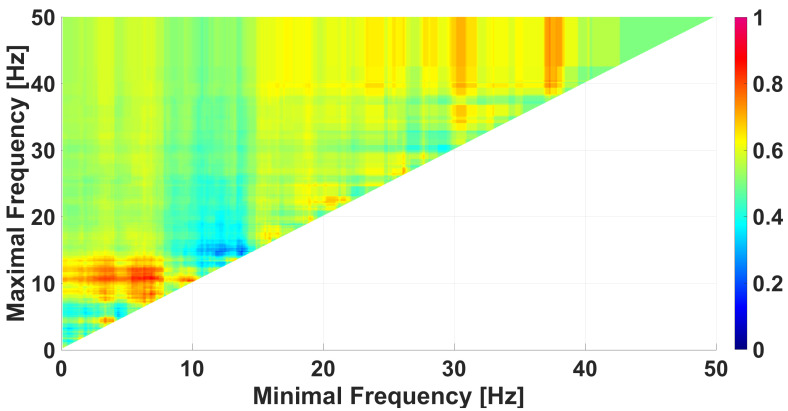
A frequency AUC diagram comparing immature discharges (9) with sleep spindles (20). *X* designates the lower limit of the frequency band; *y* designates the upper limit of the frequency band. The diagram does not contain any pronounced monochromatic areas, which indicates the difficulty of distinguishing immature discharges from sleep spindles.

**Figure 14 diagnostics-15-00983-f014:**
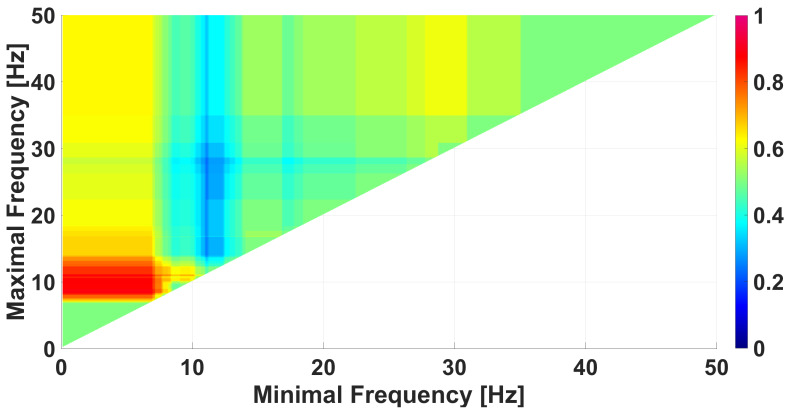
A frequency AUC diagram comparing immature discharges (9) with sleep spindles (20). The diagram is constructed using the following constraints on the wave-train parameters: duration of at least 1.6 periods, bandwidth of at least 1.7 Hz, and instantaneous phase −π up to +1.5 radians. *X* designates the lower limit of the frequency band; *y* designates the upper limit of the frequency band. One red region is observed in the diagram.

**Figure 15 diagnostics-15-00983-f015:**
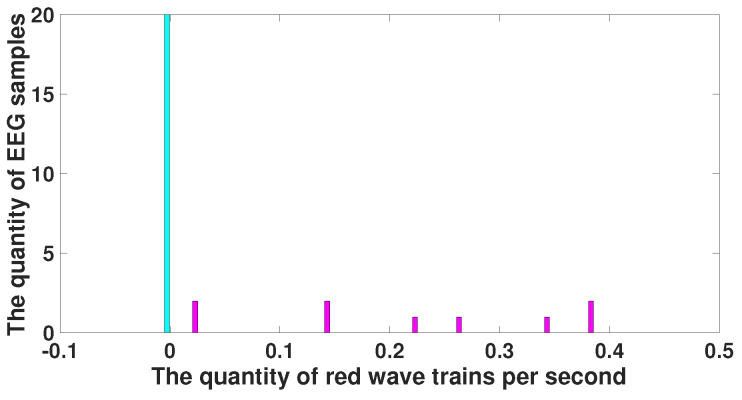
Histogram of the number of red wave trains. *X* designates the number of red wave trains per second; *y* designates the number of EEG fragments. The histogram of the number of red wave trains observed in immature discharges is drawn in purple; the histogram of the number of red wave trains observed in sleep spindles is drawn in light blue. The purple and blue histograms partially intersect.

**Figure 16 diagnostics-15-00983-f016:**
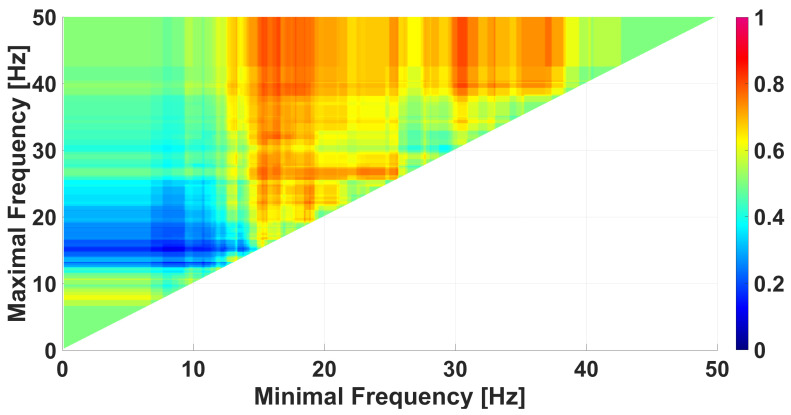
A frequency AUC diagram comparing immature discharges (9) with sleep spindles (20). The diagram is constructed using the following constraints on the wave-train parameters: a bandwidth of at least 4 Hz and an instantaneous phase from −2.5 to +2.5 radians. *X* designates the lower limit of the frequency band; *y* designates the upper limit of the frequency band.

**Figure 17 diagnostics-15-00983-f017:**
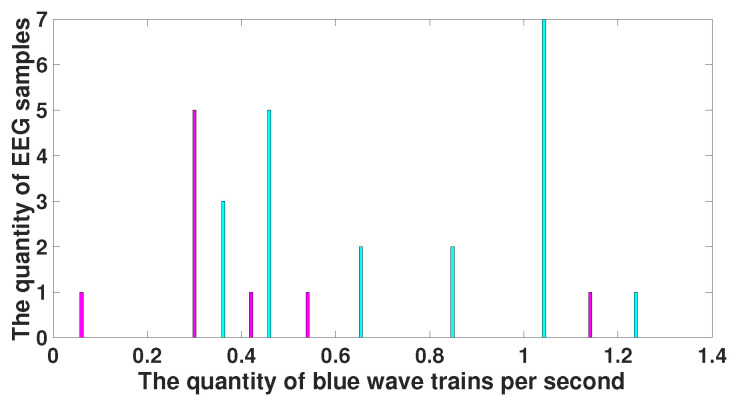
Histogram of the number of blue wave trains. *X* designates the number of blue wave trains per second; *y* designates the number of EEG fragments. The histogram of the number of wave trains observed in immature discharges is drawn in purple; the histogram of the number of wave trains observed in sleep spindles is drawn in light blue. The purple and light blue histograms partially intersect.

**Figure 18 diagnostics-15-00983-f018:**
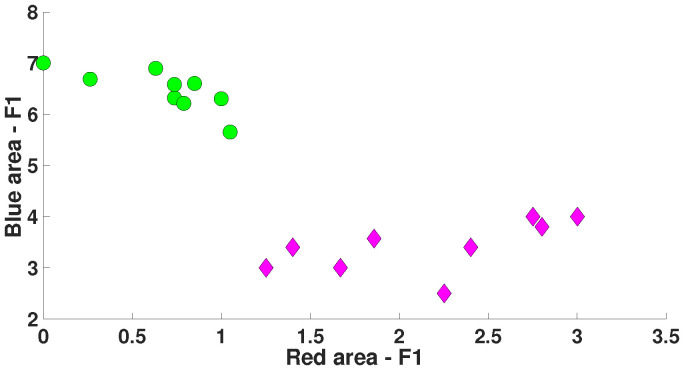
A scatter plot of the number of wave trains in immature discharges (9) and background EEG (9) in the blue and red areas of the AUC diagrams (see Table 1). *X* designates the number of wave trains per second in the red areas of the AUC diagrams. *Y* designates the number of wave trains per second in the blue areas of the AUC diagrams. Purple diamonds designate immature discharges. Green circles designate fragments of the background EEG.

**Figure 19 diagnostics-15-00983-f019:**
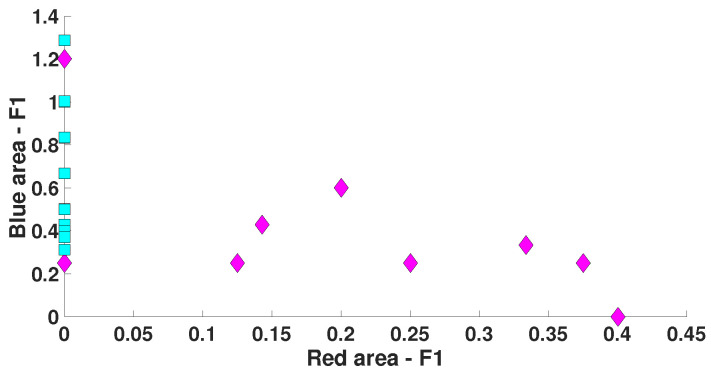
A scatter plot of the number of wave trains in immature discharges (9) and sleep spindles (20) in the blue and red areas of the AUC diagrams (see Table 2). *X* designates the number of wave trains per second in the red areas of the AUC diagrams. *Y* designates the number of wave trains per second in the blue areas of the AUC diagrams. Purple diamonds designate immature discharges. Blue squares designate sleep spindles.

**Table 1 diagnostics-15-00983-t001:** Parameters of wave trains in EEG wavelet spectrograms that distinguish immature discharges (9) from background EEG (9).

Parameter	Blue Area	Red Area	Units
PSD	-	from 50,000 and above	μV^2^/Hz
Duration in seconds	-	-	s
Central frequency	-	-	Hz
Duration in periods	-	-	periods
Instantaneous phase	-	-	radians
Bandwidth	from 2 and above	-	Hz
AUC value	0	1	-
Probability of the type 1 error (*p*-value)	<0.0001	<0.0001	-

**Table 2 diagnostics-15-00983-t002:** Parameters of wave trains in EEG wavelet spectrograms that distinguish immature discharges (9) from sleep spindles (20).

Parameter	Blue Area	Red Area	Units
PSD	-	-	μV^2^/Hz
Duration in seconds	-	-	s
Central frequency	No more than 15	No more than 8.5	Hz
Duration in periods	-	from 1.6 and above	periods
Instantaneous phase	from −2.5 to +2.5	from −3.14 to +1.5	radians
Bandwidth	from 4 and above	from 1.7 and above	Hz
AUC value	0.175	0.8889	-
Probability of the type 1 error (*p*-value)	0.0062	<0.0001	-

## Data Availability

The experimental data are fully presented in this manuscript. Additional information may be provided upon a reasonable request.

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
