# Peer review of "The Diagnostic Value of EEG Wave Trains for Distinguishing Immature Absence Seizures and Sleep Spindles: Evidence from the WAG/Rij Rat Model"

_diagnostics, 2025, doi:10.3390/diagnostics15080983_

Round 1

Reviewer 1 Report

Comments and Suggestions for Authors

This is a brilliant work of great importance both from the scientific and practical points of view. Using a special genetic strain of rats predisposed to attacks of petit mal epilepsy as a model and applying modern mathematical methods of analyzing complex signals, the authors were able to identify subtle differences between absence bursts of activity on the EEG, especially immature ones, and spindles of normal non-REM sleep. The authors suggest that the developed method opens up new prospects for solving the problem of early diagnosis of absence epilepsy, especially in children. I have no comments on this work. I strongly recommend it for early publication.

Author Response

Dear Reviewer 1,

Thank you so much for appreciating our work.

Reviewer 2 Report

Comments and Suggestions for Authors

A- General Comments:
The manuscript presents an important study on distinguishing immature absence seizures from sleep spindles using EEG analysis in the WAG/Rij rat model. The topic is relevant for early diagnosis of absence epilepsy and provides valuable insights into the differentiation of EEG patterns using wave train electrical activity analysis. The study is well-structured, and the methods are rigorously applied.

Strengths:
1- Novelty & Contribution: The study provides a unique approach using AUC diagrams and wavelet spectrograms to differentiate epileptic discharges from sleep spindles.
2- Methodological Rigor: The use of both visual and statistical analyses, including genetic algorithms for parameter optimization, strengthens the study's validity.
3- Reproducibility: The well-documented methodology facilitates replication in future research.
Suggestions for Improvement:
4- Clarify Clinical Relevance: While the findings are compelling, the manuscript could benefit from a discussion on the potential translation of this method to human EEG studies for early epilepsy diagnosis.
5- Comparison with Existing Methods: A direct comparison with other automated detection methods (e.g., deep learning models or other machine learning classifiers) would strengthen the argument for the proposed method.
5- Parameter Sensitivity Analysis: Additional explanation on how different parameter choices (such as wave train frequency, bandwidth, and power spectral density thresholds) affect classification performance.
Visualization Enhancement: Some figures, particularly the AUC diagrams, could benefit from clearer labeling and color contrast to improve readability.
B- Minor Suggestions:
Ensure that abbreviations (such as SWD, PSD, and AUC) are defined at their first appearance.
Improve flow between sections, particularly in the transition from methods to results.

Comments on the Quality of English Language

The manuscript is generally well-written and conveys the research clearly. However, there are some areas where the language could be improved for better readability and clarity.

Areas for Improvement:

Some sentences are overly complex and could be simplified for better readability.
Minor grammatical errors and awkward phrasing are present in some sections (e.g., "the diagnosis of convulsive temporal lobe epilepsy with well-marked paroxysmal spikes on the interictal EEG is much easier" could be rewritten for smoother readability).
Articles (e.g., "the," "a") are occasionally missing or misused.
Some figure captions could be more precise for better understanding.
A careful proofreading or professional English editing service is recommended to enhance clarity and readability.

Author Response

Dear Reviewer 2,

We would like to express our appreciation for your contribution to our manuscript. We sincerely hope that we were able to respond adequately to your comments and concerns. The manuscript has been revised according to your comments and remarks. All necessary edits have been made. All figures have been corrected in accordance with your comments. Three references have been added to the list of references. We hope that we succeeded in improving the weak points of our paper.

4 - Clarify Clinical Relevance: While the findings are compelling, the manuscript could benefit from a discussion on the potential translation of this method to human EEG studies for early epilepsy diagnosis.

Thank you for your recommendation. In the revised version of our manuscript, we discussed the potential translation of the developed method for human EEG analysis. In particular, we have indicated that “Criteria have been formulated that can be used for automatic recognition of immature epileptic activity and sleep spindles, which may have translational significance for clinical use. The wave train EEG analysis developed on the WAG/Rij rat absence model can be used to analyze human EEG for early diagnosis of epilepsy”.

5 - Comparison with Existing Methods: A direct comparison with other automated detection methods (e.g., deep learning models or other machine learning classifiers) would strengthen the argument for the proposed method.

Thank you for your valuable recommendation. This manuscript presents a new method for extracting features necessary for the detection/recognition of immature absence seizures and their differentiation from sleep spindles. At the next stage of our research, we plan to apply this method to automatically detect immature epileptic activity. Altogether, it can be concluded that the proposed method for extracting the features of immature epileptic activity has an advantage over deep learning models. This advantage is that the proposed method does not require a large amount of data and can operate with a small dataset.

6 - Parameter Sensitivity Analysis: Additional explanation on how different parameter choices (such as wave train frequency, bandwidth, and power spectral density thresholds) affect classification performance.

Thank you for this useful comment. Parameter Sensitivity Analysis is a very important and interesting, but special task. In our next paper, we will analyze how the choice of various parameters (wave train frequency, bandwidth, and power spectral density) affects recognition efficiency.

7 - Visualization Enhancement: Some figures, particularly the AUC diagrams, could benefit from clearer labeling and color contrast to improve readability.

Thank you for these remarks. The figures have been improved. The elements of the diagrams are enlarged and the color contrast is improved.

8 - Ensure that abbreviations (such as SWD, PSD, and AUC) are defined at their first appearance.

In accordance with your remark, abbreviations have been defined at their first appearance.

9 - Improve flow between sections, particularly in the transition from methods to results.

Thank you for your remark. The flow between sections, particularly in the transition from methods to results, has been improved.

10 - The manuscript is generally well-written and conveys the research clearly. However, there are some areas where the language could be improved for better readability and clarity.

Thank you for this remark. We will request the English proofreading to improve the language for better readability of the manuscript.

Finally, we would like to appreciate you once again for your attentive attitude to our work and useful comments that allowed us to improve our paper.

Round 2

Reviewer 2 Report

Comments and Suggestions for Authors

The manuscript in it's form looks great. I recommend accpetance.